# Economic assessment of precautionary measures against floods: insights from a non contextual approach

Richert, Claire [1], Boisgontier, Hélène [1], and Grelot, Frédéric [1]

[1]UMR G-EAU, Irstea, Université de Montpellier

**Correspondence:** Claire Richert (claire.richert@irstea.fr)

**Abstract.** To limit the losses due to floods, public authorities can try to foster the adoption of private measures aimed at reducing the vulnerability of dwellings. However, the efficacy and cost-efficiency of such measures to reduce material losses are not well-known. In particular, the influence of building and flood characteristics on these variables has not been thoroughly studied. A better understanding of this topic would help identify the measures that are relevant to implement in specific contexts. To address this gap, we examined the effect of building and flood characteristics on the cost, efficacy, and cost-efficiency of three groups of measures taken on existing dwellings: one consists in elevating the dwelling, one in dry-proofing it, and one in using construction materials that are resistant to water or cheap to repair or replace. We combined expert judgement and computer modelling to assess their cost, efficacy, and cost-efficiency for a wide range of flood depth and duration, building characteristics, and level of exposure. We found that the value of the building components has a positive effect on the efficacy of dry-proofing and elevating a dwelling. Both the efficacy and cost of these two groups of measures increase with the size of the dwelling. Moreover, according to our results, dry-proofing and elevating a dwelling are unlikely to be cost-efficient for dwellings that are not exposed to floods with a return period lower than 100 and 30 years, respectively. Our findings also highlight that it is often less expensive to use the adapted than the original materials when rebuilding a damaged dwelling. Moreover, adapting the materials of an intact dwelling is unlikely to be cost-efficient for dwellings that are not exposed to floods with a return period lower than 20 years. Our results apply to France because the damage and the installation costs of the measures are specific to France and the geometry of the dwellings considered to perform our analyses is based on French dwellings.

## 1 Introduction

A flood risk can be defined as the combination of a hazard, exposed assets and populations, and their vulnerability to the hazard (e.g. Apel et al. (2009)). Thus, mitigating such a phenomenon boils down to reducing at least one of these three components. Throughout the twentieth century, it was largely preferred in several countries to only target the hazard component of flood risks through the building of physical defences (e.g. in the Netherlands (Vis et al., 2003), in the United Kingdom (Werritty, 2006), or in the United States (Tobin, 1995)). Nowadays, several authors report that flood risk management policies are increasingly combining flood defences with measures aimed at reducing people and assets' vulnerability (Klijn and Samuels, 2008; Merz et al., 2010). This tendency is driven by the awareness that physical defences can harm ecosystems and that levee failures

can have catastrophic consequences (Klijn and Samuels, 2008), and by the increasing uncertainty regarding the risks of floods due to climate change (Merz et al., 2010). Thus, in countries such as England, France, Hungary, or Germany, recent policies were designed to limit development in flood prone areas, provide more efficient insurance schemes, raise people's awareness of the risk, or foster the adoption of measures aimed at reducing the vulnerability of dwellings (Klijn and Samuels, 2008).

Policy-makers need to be able to compare these different means of reducing the vulnerability to floods in order to efficiently allocate public funds among them, in terms of subsidies and communication.

We focused on measures aimed at reducing the vulnerability of existing dwellings to floods, which we call "precautionary measures", and analysed their efficacy, cost, and cost-efficiency. We define the cost-efficiency of a measure as the discounted sum of the difference between its annual expected efficacy and its annual cost over its lifespan. The efficacy of a precautionary

measure indicates the extent to which it reduces the level of damage to a dwelling. It depends on the flood intensity. The annual expected efficacy is thus the probability-weighted average of the values of efficacy computed for all possible flood intensities.

So far, the efficacy of precautionary measures has been examined at the household level using mainly empirical approaches and expert judgement (see Kreibich et al. (2015) for a review). For instance, the International Commission for the Protection of the Rhine relied on expert judgement to assess the efficacy of flood proofing measures (International Commission for the

Protection of the Rhine (ICPR), 2002). On the other hand, Kreibich et al. (2005) surveyed people who were affected by the flood that took place in 2002 in the Elbe river basin. They compared the damage to the building and contents of dwellings with and without different types of precautionary measures. Their results suggest that installing flood barriers and adapting the building structure, use and layout to floods were effective measures to reduce material damage in their case study. The same conclusion was reached by Kreibich and Thieken (2009) after comparing the levels of damage due to different floods

that occurred in a given area in Germany and between which the proportion of inhabitants who took measures had increased. Using a similar approach on another case study in Germany, Bubeck et al. (2012) also suggest that precautionary measures significantly reduced material damage.

All these studies assessed the efficacy of precautionary measures for particular flood events. However, the comparison of the cost and efficacy of a measure requires to estimate its annual expected efficacy. Kreibich et al. (2011); Poussin et al. (2015);

Xian et al. (2017) proposed various methods to assess the annual expected efficacy of some measures. Kreibich et al. (2011) surveyed people who were affected by floods in Germany. They assumed that the mean efficacy of a measure is the difference between the mean amount of damage suffered by those who lived in houses where the measure was not implemented and the mean amount of damage suffered by the others. They multiplied the estimated mean efficacies by several frequencies to obtain annual expected efficacies that relate to various flood return periods. Poussin et al. (2015) conducted a survey in France to

assess the cost-efficiency of some precautionary measures. They used ordinary least square regression models to explain the relationship between the amount of damage and several independent variables, such as the presence of some precautionary measures and the flood depth. They assumed that the mean efficacy of a measure is the difference between the estimated levels of damage suffered by an average home without and with this measure. To assess the annual expected efficacy, they proceeded in the same way as Kreibich et al. (2011). Xian et al. (2017), assumed that the annual expected efficacy of elevating a dwelling

is the reduction in risk-based annual insurance premium due to this measure, which they computed following the guidelines of

the Federal Emergency Management Agency (FEMA, 2014, 2018). As explained in a report of the National Research Council (2015), the risk-based annual insurance premium takes into account the average annual expected damage, which is obtained across classes of buildings which are in the same flood zone and share some characteristics, including their elevation. The values of average annual expected damage are computed using two types of depth-damage functions: some were obtained by using claims data and others come from the U.S. Army Corps of Engineers but the method used to obtain them is not well-documented, according to the National Research Council (2015). Thus, Xian et al. (2017) ultimately relied partly on empirical data and partly on data of unknown nature to assess the annual expected efficacy of elevating a dwelling.

While empirical studies analyse precautionary measures in realistic settings, the results they provide are largely context-dependent, as reported by Poussin et al. (2015) and Kreibich et al. (2015). Moreover, since they do not control for all the parameters that influence the amount of damage, they cannot be used to anticipate the efficacy or cost-efficiency of precautionary measures in other contexts. In particular, they do not take into account the relationship between the materials used for the components of the buildings and the vulnerability to floods of these latter, which could partly explain the variance in the amount of damage (National Research Council, 2015). As for the reports based on expert judgement, since they contain little methodological information, it is difficult to evaluate their reliability in specific contexts.

In brief, the existing literature focuses on assessing the efficacy or cost-efficiency of precautionary measures, rather than on explaining their variability. The aim of our study was to address this gap. We combined data based on expert judgement and computer modelling to analyse three types of measures (elevation, dry-proofing, and component adaptations) for a wide range of flood intensities and dwellings characteristics, including the materials used for their components. More specifically, we assessed ranges of cost and efficacy of the measures and examined the influence of building and flood characteristics on these variables. For each type of measures, we also found a range of exposure level for which it is unlikely that the measure could be cost-efficient, independently of the building characteristics.

In the following section, we present the measures that we focus on. We describe the method used to assess their efficacy, cost, and cost-efficiency in Sect. 3. Then, we present the results in Sect. 4. We discuss them in Sect. 5 and conclude in Sect. 6.

## 2 Precautionary measures studied

We reviewed the measures recommended to reduce the vulnerability of dwellings to floods in a joint report of the French ministries in charge of environment and housing (Ministère de l'égalité des Territoires et du Logement - Ministère de l'écologie, du Développement durable, et de l'énergie, 2012), a report of the European Center for Flood Risk Prevention (CEPRI, 2010), and a report of the Doubs and Saône catchment management agency (EPTB Saône et Doubs, 2015).

The measures described in these reports can be classified into three categories: some aim at avoiding damage by elevating the house, storing valuables upstairs, or elevating heating and electrical utilities for example, some at preventing the water from entering into the dwelling, and some at limiting the costs of repair and replacement of building components.

We analysed the measure that consists in elevating the house, and those that aim at preventing the water from entering into the dwelling or at limiting the costs of repair and replacement of the building components. We refer to the first measure as

the elevation strategy, to the second group of measures as the dry-proofing strategy, and to the third group of measures as the component adaptations strategy.

## 2.1 Elevation

To avoid damage up to a given flood depth, dwellings can sometimes be elevated. According to a report of the FEMA (2009), if a dwelling is built on a basement, a crawlspace, or on open foundations, the first step to elevate it consists generally in separating it from its foundations and raising it on hydraulic jacks while it is held by a temporary support. Then, the existing foundations can be extended or new ones can be built. If the dwelling has slab-on-grade foundations, they are lifted together and new foundations are constructed below the slab.[1] In both cases, an external staircase must be built to access the dwelling and utility lines must be extended.

We analysed the efficacy, cost, and cost-efficiency of elevating dwellings by 50 cm, 100 cm, and 250 cm. We assumed that only single storey houses can be elevated because other types of dwellings are too large.

## 2.2 Dry-proofing

Measures can be taken to prevent the water from entering into a dwelling if the flood depth stays below 1 m and the flood lasts less than 48 hours. For higher flood depths or longer flood durations, the pressure on the vertical elements of the building structure can cause severe damage.

For a dwelling without a basement, the following measures must be taken together and dimensioned consistently to prevent the water from entering up to a chosen threshold: installing flood barriers, repairing faulty seals in the external walls, water-proofing the external walls, treating cracks in the external walls, installing removable covers on small openings that are below the chosen threshold, installing anti-backflow valves, ensuring that the electrical cable sleeves are watertight, and buying a pumping device.[2] Following Poussin et al. (2012), we call "dry-proofing" the combination of these measures. We assessed the efficacy, cost, and cost-efficiency of dry-proofing a dwelling without a basement in the cases where it prevents the water from entering up to flood depths of 50 cm and 1 m.

## 2.3 Component adaptations

We call "component adaptation" a measure which consists in using building materials that are resistant to water or cheap to repair or replace. We studied together component adaptations which pertain to the ceilings, the walls, the floors, and the openings. We detail in Table 1 the component adaptations that pertain to each of these building components.

---

[1]It is also possible to leave the slab on the ground and to construct a new floor after lifting the house.

[2]For dwellings with a basement, additional measures must be taken. We only analysed the measures which aim to prevent the water from entering into a dwelling without a basement.

## 3 Method

### 3.1 Overview

We used a computer tool called *floodam* (Grelot and Richert, 2019) and developed in R language (R Core Team, 2017) to assess the efficacy of the strategies aimed at reducing the vulnerability of existing dwellings. *floodam* requires as input a numerical model of a building and produces a function which associates a level of damage to several combinations of flood depth and duration. We call this type of output a damage function. *floodam* has been developed to produce national damage functions that are recommended by the French State in its cost-benefit analysis methodology (Rouchon et al., 2018).

The efficacy and the cost of the strategies were assessed for several numerical models of dwellings. The cost of each strategy depends on the geometry of the dwellings and, in some cases, on whether the strategy is taken on an intact or damaged building and on the building materials. The efficacy is assessed by comparing the damage functions of the dwellings obtained with and without a strategy. It depends on the flood depth and duration, on the geometry of the dwellings, and on the building materials.

In this section, we describe the numerical models used as inputs, we provide an overview of how *floodam* works, we present the advantages of *floodam* to assess the strategies compared to other flood damage models, and we describe how we modelled the strategies. We then explain how we assessed the cost, the efficacy, and the cost-efficiency of each strategy.

### 3.2 Numerical models of dwellings

The numerical model of a dwelling is made up of an xml file and a csv file. The former indicates the ground floor height above ground level, the layout and size of the rooms, and the construction materials used for the building. Figure 1 provides a graphical interpretation of the xml file of a numerical model. The csv file indicates the pieces of furniture that are present in each room and their height above the floor. For instance, Table 2 shows the information that pertains to a bedroom contained in the csv file of a numerical model.

Originally, we had three numerical models which represent real dwellings: we visited an apartment to establish its plan and make an inventory of its furniture, the European Center for Flood Risk Prevention did the same with a single storey house (CEPRI, 2014) and gave us access to their data, and we used the architectural plan of a double storey house. Since we had no real data regarding the furniture of the double storey house, we first allocated the pieces of furniture of the single storey house in the corresponding rooms of the double storey house. Then, we added pieces of furniture in one room which had no correspondence in the single storey house (a home office) and in two dressing rooms. More specifically, we assumed that the home office contained a desk, a chair, and a cupboard in chipboard and that the dressing rooms each contained a variety of small items.

In order to be able to generalize our results, we developed several versions of these three numerical models of dwellings by modifying the combination of the building components listed in Table 3. These components were chosen because they are widely used in French buildings according to a report of the agency of building quality (Agence Qualité Construction, 2009). Each version contains only one variant of each component listed in Table 3. Since there are 1 728 possible combinations of all

components, we developed the same number of versions for each of the three original numerical models of dwellings. In total, we thus have 5 184 numerical models of dwellings.

## 3.3 Overview of *floodam*

We define the damage suffered by a good as the expected cost of the actions that must be performed after a flood in order to
bring it back to its pre-flood state. Using this definition, *floodam* relies on two main assumptions: 1) the damage suffered by a building is the sum of the levels of damage suffered by its components, and 2) the levels of damage of the components are independent.

*floodam* relies on a database of 431 elementary damage functions. An elementary damage function associates a level of damage suffered by an elementary component to several combinations of flood depth and duration. A partition wall in plaster
or a washing machine are examples of elementary components. The elementary damage functions come from interviews with insurance and construction experts. We used elementary damage functions with data points at immersion depth values from 0 to 500 cm (included) in 10 cm increment, and immersion duration values from 0 to 144 hours (included)[3] in 12 hours increment.

For a given numerical model, *floodam* computes the height above the ground of all elementary components and the size or quantity of the elementary components of the building. Then, for each combination of depth and duration of immersion,
it 1) computes the depth of immersion of each elementary component, 2) retrieves the corresponding levels of damage by measurement unit for the duration of immersion considered, 3) multiplies each level of damage by the size/quantity of the corresponding elementary component, 4) sums the obtained values to obtain the level of damage at the scale of the dwelling. The damage function of the numerical model considered is made up of the levels of damage computed for each combination of immersion depth and duration.

*floodam* is used to estimate the damage due to floods that do not cause failure of walls and that do not involve salt water.

A more detailed description of *floodam* can be found in Grelot and Richert (2019).

## 3.4 Suitability of *floodam* to assess the precautionary measures

Numerous empirical and synthetic flood loss models exist. *floodam* belongs to the latter category. Empirical models are based on observed flood loss data, whereas synthetic models rely on a description of flood damage mechanisms (Gerl et al., 2016).
Some empirical flood damage models include precautionary measures as explanatory variables (see for example Kreibich et al. (2017)) and can be used to estimate their mean efficacy (Sairam et al., 2019). These models account for the mean effect on flood damage of all the measures observed in the case studies used to produce them. Thus, they cannot be used to estimate the efficacy of specific precautionary measures. Moreover, the influence of the flood parameters and building characteristics on the efficacy of precautionary measures cannot be deduced from these models.

---

[3]Some floods can last more than 144 hours. The experts interviewed did not provide data regarding the vulnerability of the elementary components to such floods. Thus, we did not study the efficacy of precautionary measures for floods longer than 144 hours.

The damage mechanisms are more explicit in synthetic models (see for example Custer and Nishijima (2015); Dottori et al. (2016); Nadal et al. (2010); Zevenbergen et al. (2007)). They can be altered to depict the effect of specific precautionary measures.

To our knowledge, *floodam* is the synthetic model based on the most detailed database of elementary damage functions. This characteristic enabled us to examine the influence of a wide variety of building materials on the efficacy of specific precautionary measures.

### 3.5 Modelling of the strategies

#### 3.5.1 Elevation

Modelling the elevation of a dwelling consists in choosing a threshold of flood depth below which damage should be avoided. The cost and efficacy of the strategy are deduced from this threshold.

#### 3.5.2 Dry-proofing

To model the dry-proofing of a dwelling, a threshold below which the water should be prevented from entering must first be chosen. The number of openings that are below the threshold is the number of flood barriers that must be installed. Then, the perimeter of the dwelling, from which the quantity of removable covers that must be installed is deduced, is computed. The perimeter is multiplied by the threshold to obtain the area on which faulty seals and cracks in the external walls must be repaired, and on which the external walls must be waterproofed.

#### 3.5.3 Component adaptations

Adapting a given component of the numerical model of a dwelling boils down to replacing its original variant by the recommended one (cf Table 1) if the latter is different from the former.

### 3.6 Analysis

#### 3.6.1 Assessment of the cost

**Elevation**

To assess the cost of elevating a dwelling, we relied on the estimates provided by the FEMA (2009) and reported in Table 4. We first converted each of these estimates in 2017 Euros by square foot. To do so, we multiplied them by the ratio between the construction prices in France and the United States (0.97, cf https://www.fgould.com), the exchange rate between the US dollar and the euro in 2009 (0.72), and an index that gives the construction price development in France between 2009 and December 2017 (1670 \((1507 + 1502 + 1498 + 1503)\4), cf https://www.insee.fr).

Once the costs were in Euros by square foot, we converted them in Euros by square meter by dividing them by $0.3048^2$ ft²/m² (1 ft = 0.3048 m). After that, we computed the cost for elevations of 50 cm, 100 cm, and 250 cm by assuming that the cost increases linearly between 50 cm and 250 cm. Finally, we used the costs indicated in Table 5.

**Dry-proofing**

The cost of each measure that must be implemented to dry-proof a dwelling was assessed by a construction expert for the single storey house. Knowing the characteristics of this dwelling, we estimated the cost of each measure by measurement unit. For each numerical model of a dwelling, they were multiplied by the quantities on which the measures must be applied. Table 6 indicates the cost of each measure and its measurement unit.

**Component adaptations**

A given component can be destroyed or intact when the adaptation takes place. At the level of a dwelling, we consider 2 situations: either all the components are destroyed or they are all intact when the adaptation takes place. In the first case, which we call the repair context, the adaptation cost is the difference between the costs of installing the recommended and the original variants of the components. In the second case, which we call the prevention context, the adaptation cost is the sum of the costs of installing the recommended variant of the components (if they are different from the original ones) and of reinstalling the 15 original coatings (of the walls, floors, and ceilings).

### 3.6.2 Assessment of the efficacy

We define the efficacy of a strategy for a given numerical model of a dwelling as the difference between the damage functions computed without and with the strategy.

**Elevation**

When a dwelling is elevated of $x$ centimetres, its damage function ($f$), which depends on the flood depth ($de$) and duration ($du$), becomes $g(de, du)$ such as :

$$g(de, du) = \begin{cases} f(de - x, du), & \text{if } de - x \geq 0 \\ 0, & \text{otherwise} \end{cases} \tag{1}$$

Therefore, the efficacy of elevating a given numerical model depends on the immersion depth and duration. For example, elevating the original version of the single storey house by 1 m can reduce the damage by up to €30 000 approximately, as 25 shown in Fig. 2.

**Dry-proofing**

The damage function of the numerical model of a dwelling where the dry-proofing strategy is installed is equal to zero for combinations of immersion depths below or equal to the threshold and immersion durations below or equal to 48 hours. For all other combinations of immersion depth and duration, dry-proofing has no effect on the damage function. For example, Fig. 30 2 shows the damage functions of the original version of the single storey house without and with the dry-proofing strategy

with a threshold of 1 m, and the resulting efficacy. For this dwelling, the maximum avoided damage due to dry-proofing is approximately €25 000.

**Component adaptations**

The efficacy of a component adaptation to reduce the vulnerability of a dwelling is the difference between the damage functions computed with the original and recommended variants of the component. Hence, the efficacy depends on the immersion depth and duration. For instance, Fig. 2 shows that the maximum avoided damage due to the component adaptations strategy in the original single storey house is lower than €5 000.

### 3.6.3 Assessment of the maximum cost-efficiency

The maximum cost-efficiency of a strategy for a given type of dwelling (single-storey house, double-storey house, apartment) is defined as a supremum of the cost-efficiency computed for the version for which the strategy is the most cost-efficient. It is thus a supremum of the cost-efficiency for the type of dwelling considered. In other words, for a given strategy and a given type of dwelling, the cost-efficiency of the strategy is always lower than the maximum cost-efficiency, regardless of the building materials or the relationship between the flood intensity and frequency.

In this section, we mathematically define the cost-efficiency and maximum cost-efficiency of a strategy.

The cost-efficiency ($CE$) of a strategy is defined for a contextualized dwelling, which is a dwelling that has a given location, and thus a given exposure to floods depending on their frequency. The cost-efficiency is the discounted sum of the difference between the annual expected efficacy ($AEE$) and the cost of the strategy over a defined time horizon ($H$) :

$$CE(H) = \sum_{1 \leq i \leq H} \frac{AEE - AEC}{(1+r)^i} - IC \tag{2}$$

where $r$ is the discount rate, $AEC$ the annual expected maintenance cost of the strategy, and $IC$ its installation cost. We assume that $AEE$ and $AEC$ are constant over the time horizon considered. A strategy is cost-efficient for a given contextualized dwelling if $CE > 0$, that-is-to-say if :

$$\frac{IC}{AEE - AEC} < \sum_{1 \leq i \leq H} \frac{1}{(1+r)^i} \tag{3}$$

Moreover, the annual expected efficiency of a strategy for a given dwelling is equal to:

$$AEE = \int_0^1 f \times E(de(f), du(f)) df \tag{4}$$

with $f$ the flood frequency, $de$ the immersion depth, $du$ the immersion duration, and $E$ the efficacy.

What we call maximum cost-efficiency is in fact an upper boundary of the cost-efficiency. We first computed it for each version of a given type of dwelling.

To compute the maximum cost-efficiency of a strategy $s$ for a given dwelling $d$, we use a lower boundary of the cost by taking only the installation cost into account. We also define the following upper boundary of annual expected efficiency:

$$AEE_{max}^{s,d}(f_{max}) = f_{max} \times E_{max}^{s,d} \tag{5}$$

$$= \frac{E_{max}^{s,d}}{T_{min}} \tag{6}$$

with $f_{max}$ the frequency of the flood that affects $d$ the most often, $T_{min}$ the return period of the flood that affects $d$ the most often, and $E_{max}^{s,d}$ the highest value of efficacy of $s$ for $d$ over all possible combinations of immersion depth and duration.

Thus, we define the maximum cost-efficiency of $s$ for $d$ as follows:

$$CE_{max}^{s,d}(T_{min}, H) = \frac{E_{max}^{s,d}}{T_{min}} \sum_{1 \leq i \leq H} \frac{1}{(1+r)^i} - IC^{s,d} \tag{7}$$

The maximum cost-efficiency of a strategy $s$ for a given type of dwelling is then the cost-efficiency computed for the version of the dwelling for which the ratio $\frac{IC^{s,d}}{AEE_{max}^{s,d}}$ is the lowest.

We used a discount rate of 2.5%, which is the value recommended to assess public investments in France (Commissariat général à la stratégie et à la prospective, 2013).

For each type of dwelling, we computed the maximum cost-efficiency for values of $H$ from 1 to 50 (for dry-proofing and component adaptations) or 100 years (for elevation) in 1 year increment and $T$ from 1 to 120 years in 1 year increment. $H$ can be considered as the lifespan of the strategy.

For each strategy and type of dwelling, we searched for the combinations of time horizon and return period for which the maximum cost-efficiency is negative. In these contexts, our results suggest that the strategy is unlikely to be cost-efficient. Indeed, unlike the cost-efficiency, the maximum cost-efficiency for a given type of dwelling does not depend on the building materials and on the relationship between the flood intensity (immersion depth and duration) and frequency. It only depends on the time horizon and return period. Thus, for the combinations of time horizon and return period associated to a negative maximum cost-efficiency, the strategy is always cost-inefficient, regardless of the building materials and on the relationship between the flood intensity and frequency.

## 4  Results

We present the ranges of cost and efficacy and the maximum cost-efficiency of the elevation, dry-proofing, and component adaptations strategies.

## 4.1 Range of cost

As shown in Table 7, the cost of a given strategy does not depend on the immersion depth and duration. It always varies with the type of dwelling and the characteristic of the strategy (the height of elevation, the threshold, or the adaptation context). In the specific case of the component adaptations strategy, it also depends on the original variant used for the building components.

The cost of elevating a single storey house lies between k€66 and k€109. Thus, it is always higher that the highest value of efficacy for this strategy. It increases with the elevation and is always the highest for dwellings that have slab-on-grade foundations.

The cost of dry-proofing a dwelling ranges from k€6 to k€10. For a given threshold, the cost of dry-proofing is always maximum for the single storey house and minimum for the double storey house. This is due to the fact that the single storey house has the greatest perimeter (54 m) and the double storey house the smallest (40 m). The area on which some measures must be applied increases with the threshold. Thus, the cost of dry-proofing is greater when the threshold is 100 cm than when it is 50 cm.

Regarding the adaptation of all building components, if it takes place on a damaged building (repair context), it is often less expensive to adapt the dwelling than to install again the original variants of the components. More precisely, it is the case for 91% of the 5 184 numerical models of dwellings. On the opposite, if the dwelling is intact when the adaptation takes place (prevention context), the mean cost of adapting all components is approximately four times higher than the cost of dry-proofing. In the prevention context, the minimum costs relate to versions of the dwellings for which the original variants of the components are highly similar to the recommended ones. Note that we did not compute numerical models of dwellings that were made up of all the recommended variants of the components. The cost and efficacy of adapting all building components of such dwellings would be null. The highest adaptation costs relate to versions of the dwellings for which almost all components must be adapted.

## 4.2 Range of efficacy

Figure 3 shows the range of efficacy obtained for each strategy. The efficacy of a strategy depends on the type and components of the dwelling, on the immersion depth and duration, and sometimes on some characteristics of the strategy.

More specifically, the efficacy of elevating a dwelling increases with the value of elevation and with the value of the components of the dwelling. For this strategy, the highest efficacy is obtained for numerical models of dwellings which contain a lot of wooden components (joist boards, parquet, and opening frames and shutters in wood). The highest efficacy is observed for an immersion depth equal to the elevation value.

As for dry-proofing, its efficacy increases with the threshold, the value of the components of the dwelling, and the floor area. The highest maximum efficacies relate to the single storey house and the lowest to the double storey house because the former has the largest floor area (133.5 m$^2$) and the latter the smallest (98 m$^2$). For a given numerical model of a dwelling, the efficacy of dry-proofing increases up to an immersion depth equal to the threshold and an immersion duration of 48 hours. In keeping

with the assumptions used when modelling dry-proofing, the efficacy is equal to zero for higher immersion depths and longer immersion durations.

Adapting all components can sometimes generate the same or a higher level of damage than keeping all the original components. This is the case for the 576 versions of each type of dwelling which originally have masonry internal walls and tiles or textile as coatings of floors. There is indeed a probability greater than zero that the adapted walls in plaster must be replaced for immersion depths greater than or equal to 30 cm, no matter the immersion duration. On the contrary, the probability that masonry walls must be replaced for immersion durations lower than 72 hours is equal to zero. This type of walls needs only to be repaired in such cases. As a consequence, for some immersion depths, the elementary damage function of an internal wall in plaster is above the one of a masonry internal wall for immersion durations lower than 72 hours. Negative levels of efficacy are observed only for dwellings which originally have tiles or textile as coatings of floors because the high efficacy of replacing parquet floors by sealed tiling floors compensates the negative efficacy of replacing masonry walls by walls in plaster. The efficacy of adapting all components is always positive for immersion durations higher than or equal to 72 hours. The variance of the efficacy increases with the quantity of building components to adapt.

Table 8 shows the distribution of the efficacy of each strategy. The efficacy of elevating a dwelling lies between €0 and k€65. Dry-proofing a dwelling leads to a reduction of damage comprised between €0 and k€36 and the efficacy of adapting all the components of the building is between k€-14 and k€54. These results highlight the high variability of the efficacy of each strategy.

## 4.3 Maximum cost-efficiency

Figure 4 indicates the maximum cost-efficiency of each strategy, depending on the type of dwelling and on some characteristics of the strategy (the elevation, the threshold, or the adaptation context), and for several combinations of time horizon and return period.

For the analysed values of time horizon, it is never cost-efficient to elevate a single storey house which is only exposed to floods with a return period higher than 30 years. If the dwelling has slab-on-grade foundations, the minimum return period for which it could be cost-efficient to elevate a dwelling is 20 years approximately.

Similarly, according to our results, adapting all building components is never cost-efficient for intact dwellings that are not exposed to floods that have a return period of less than 20 years. However, when component adaptations take place on a damaged dwelling, our results do not indicate ranges of time horizon and return period for which this strategy is never cost-efficient.

Regarding dry-proofing, the results are similar for the three types of dwellings and are not affected by the threshold. They suggest that dry-proofing is never cost-efficient for dwellings that are not exposed to floods with a return period of a hundred years or less.

Except in the case of adapting all the building components of a damaged building, we observe that the maximum return period for which a strategy is cost-efficient increases with the time horizon. For instance, for a dwelling that must be entirely

dry-proofed again after twenty years and that is not exposed to floods with a return period of less than 60 years, this strategy will never be cost-efficient.

## 5   Discussion

We assessed the cost and efficacy of some precautionary measures by taking into account some characteristics of the dwellings (their building components and size), parameters of the measures, grouped in strategies (their dimension or implementation context), and flood characteristics (immersion depth and duration). Then, we computed the maximum cost-efficiency of each strategy for several combinations of time horizon and return period. We could thus identify exposure levels for which it is unlikely that the strategies could be cost-efficient.

### 5.1   Main results

The value of the building components by square meter has a positive effect on the efficacy of dry-proofing and elevating a dwelling while it does not affect the cost of these strategies. Hence, the more expensive the components of a dwelling, the more relevant it can be to elevate or dry-proof it. By contrast, both the efficacy and the cost of dry-proofing and elevation increase with the flood depth below which damage must be avoided and with the size of the dwelling. Consequently, these parameters do not affect the maximum cost-efficiency of dry-proofing and elevation. According to our results, these strategies are unlikely to be cost-efficient for dwellings only exposed to floods with a return period higher than 100 and 30 years, respectively.

The efficacy of adapting the building components strongly depends on their original materials. It can even be negative for floods that last less than 72 hours if the internal walls are originally in masonry. The cost of adapting the building components is influenced by the adaptation context and by the original materials. If the adaptation takes place on an already damaged building, it is most of the time less expensive to adapt it than to reinstall the original variants of the components. However, it costs approximately €40 000 on average to adapt an intact building. Since the cost and efficacy both increase with the quantity of components to adapt, the maximum cost-efficiency does not depend on the size of the dwelling. It is unlikely that adapting an intact dwelling could be cost-efficient if this latter is not exposed to floods with a return period lower than 20 years. In a repair context, we could not identify exposure levels for which it is never cost-efficient to adapt all building components.

Note that the cost-efficiency of all strategies increases with their lifespan and with the level of exposure in terms of frequency of floods.

### 5.2   Comparison with previous studies

#### 5.2.1   Elevation

According to Poussin et al. (2015), the mean efficacy of elevating the ground floor is €8 000 approximately and the cost of this strategy for existing buildings lies between €25 000 and €69 000. The range of cost comes from an article written by Aerts et al. (2013). These authors also used the data from the report of the FEMA (2009) to estimate the costs of elevating

dwellings. Thus, our estimates of the cost of elevation are of the same order of magnitude as those reported by Poussin et al. (2015), even if they are higher on average (€87 000). Moreover, the mean efficacy estimated by these authors lies in the range of efficacy that we found (between €0 and €65 000). Poussin et al. (2015) also found that elevating an existing building is only cost-efficient for dwellings exposed to floods with a return period lower than 10 years. This result is compatible with our study.

Similarly to our results, those of Xian et al. (2017) indicate that the cost-efficiency of elevating a dwelling is positively affected by the value by square meter of this latter, the frequency of floods, and the lifespan of the strategy.

### 5.2.2 Dry-proofing

Zevenbergen et al. (2007) investigated the cost-efficiency of dry-proofing a dwelling until 0.9 meter. They found that it would cost €8 000 to implement this strategy on a typical Dutch single-family dwelling. This result lies within the range of cost we found for this strategy. In the two case studies examined by Zevenbergen et al. (2007), dry-proofing was cost-efficient until a return period of 30 years. This is in line with our results.

Kreibich et al. (2011) and Poussin et al. (2015) studied the efficacy, cost, and cost-efficiency of installing flood barriers.

The average efficacy of flood barriers estimated by Kreibich et al. (2011) is €23 491 and the estimated cost of this measure is €6 100 for an average house. As explained in Sect. 3, flood barriers must be complemented by other measures to dry-proof a dwelling. Since Kreibich et al. (2011) only asked whether flood barriers were installed, it is likely that the average efficacy they found is lower than the one that would have been observed for completely dry-proofed dwellings. Moreover, since the flood depth which affected the dwellings of the respondents is not known, we cannot precisely compare our results with this average efficacy. However, it lies in the range of efficacy that we found for the the dry-proofing strategy. Moreover, we found an average cost for the installation of flood barriers only of €3 100, which is two times lower than the one used by Kreibich et al. (2011) but stays in the same order of magnitude. According to the results of Kreibich et al. (2011), flood barriers are only cost-efficient for dwellings affected by floods with a return period lower than 40 years approximately. This result is compatible with ours since we found that it is unlikely that dry-proofing could be cost-efficient for dwellings only exposed to floods with a return period higher than 100 years.

Poussin et al. (2015) found that flood barriers did not significantly reduce material damage in their whole sample. However, they highlight that this result may not be reliable because of multi-collinearity issues.

### 5.2.3 Component adaptations

We did not find studies which analyse specifically the efficacy, cost, or cost-efficiency of adapting all the building components. However, Kreibich et al. (2005) studied the efficacy of the combination of using waterproof building materials and having mostly movable pieces of furniture on the ground floor. They found that this strategy reduced the total damage by €39 000 on average. In line with this result, our estimates of the efficacy of adapting all building components are comprised between €-14 000 and €54 000.

According to Poussin et al. (2015), adapting the walls and equipment increases the damage by €2 000 on average. This result is compatible with our finding that adapting the walls can lead to negative levels of efficacy. These authors did not estimate the cost of this measure. They also found that adapting the floors could reduce the damage by €400 to €10 000. They used a cost for this measure that lies between €800 and €7 250. They do not specify the adaptation context for which this cost was estimated, but given that it is always positive, we assume that it was assessed for an intact building. To adapt only the floors, we found costs in a prevention context comprised between €10 000 and €23 000. These values are higher than those used by Poussin et al. (2015). As for the efficacy of adapting only the floors, we estimate that it lies between €0 and €16 000 depending on the flood depth and duration. These values are in line with those found by Poussin et al. (2015).

## 5.3  Recommendations based on our results

Our results seem reliable since they are mostly in line with previous studies. They suggest that elevating or adapting the building components of intact dwellings that are not exposed to frequent floods (i.e. with a return period lower than 30 years) should not be fostered by policy-makers who wish to limit material damage due to floods. However, after a flood, it could be efficient to take advantage of the reconstruction phase to adapt the building components, since it is often less expensive to install the recommended components than to rebuild dwellings as they were before the event. Given that the post-disaster recovery often occurs in a climate of urgency, it could be useful to design in advance policy tools to help people adapt their dwelling during this phase.

Moreover, policy-makers should not promote the installation of dry-proofing measures in dwellings that are not exposed to floods with a return period lower than 100 years.

Besides the level of exposure, the building components should be taken into account to assess the vulnerability of the dwellings, and thus the relevance of implementing precautionary measures that sometimes generate higher costs than benefits.

## 5.4  Limits

Decisions regarding the allocation of public funds to communicate about the strategies considered here or to subsidize their installation cannot be based solely on the present study because of several limits relating either to its perimeter, or to the method on which it relies. Several limits due to the method are linked to the assumptions and data used in *floodam*.

### 5.4.1  Limits linked to the study perimeter

We only took into account the damage in terms of monetary losses and did not consider the impact of the strategies on the damage related to human health. While this latter is unlikely to be influenced by component adaptations, it could be reduced by dry-proofing and even more by elevating the dwelling.

Moreover, our results cannot be used to identify a range of exposure for which it is likely that the studied precautionary measures would efficiently reduce monetary losses. Contextualised studies are required to finely relate the efficiency of precautionary measures to the characteristics of a dwelling and its level of exposure to floods.

### 5.4.2 Limits linked to the method

Our results are mainly relevant for relatively slow riverine floods. Indeed, *floodam* assumes that the pieces of furniture are never moved by the water during a flood and that the salinity level of the water is negligible.

Moreover, our results apply to France because we used French data to estimate the costs of the measures, the database of elementary damage functions of *floodam* relies on French costs, and we used French dwellings to develop the numerical models. These elements should be adapted to conduct our study in another country.

We only studied the efficacy and cost-efficiency of the strategies for dwellings that are exposed to floods that do not last more than 144 hours because the experts interviewed to develop the elementary damage functions of *floodam* did not have information about the consequences of longer floods. The efficacy of dry-proofing for such floods is null because it is recommended to let the water enter the building after 48 hours. As for the elevation strategy, its efficacy in case of floods that last more than 144 hours depends on the propensity of such floods to generate foundations failure. If the foundations fail, the fact that the building is elevated does not reduce the damage. The efficacy of the component adaptations strategy for floods longer than 144 hours depends on the vulnerability of the recommended components when they are in contact with water for more than 144 hours.

Our results also depend on the geometry of the dwellings used to develop the numerical models. For instance, dry-proofing was only analysed for dwellings that do not contain a basement.

*floodam* is the tool used by the French State to produce currently recommended national flood damage functions to be used in cost-benefit analyses. The comparison of these damage functions to formerly recommended empirical flood damage functions and to empirical data collected in the South of France (CEPRI, 2014) led to this recommendation. The national damage functions were also used in a case study in the South of France and the estimates obtained were compared with empirical flood damage data (Richert and Grelot, 2018): the mean damage estimate amounted to 70% (99%) of the mean empirical damage to houses (apartments). Nevertheless, in the present article, we have used *floodam* for a wide range of configurations for which validation against empirical data has not been done. Such validation would imply to have access to detailed data about a given flood: they should indicate precisely the location and characteristics (in terms of building materials, furniture, and geometry) of the dwellings of the flooded area, the flood depth and duration outside and inside these dwellings, and the flood damage suffered by each dwelling. At the present moment, this type of data are not available.

## 6 Conclusions

We analysed three types of precautionary measures in a non contextualized setting. This novel approach enabled us to explore the influence of several building and flood characteristics on the cost and efficacy of the precautionary measures and to find ranges of exposure for which they are unlikely to be cost-efficient. In particular, we found that adapting all the building components or elevating an existing dwelling are unlikely to be cost-efficient if the probability of occurrence of floods is lower than 1/30 per year. As for dry-proofing, this measure is unlikely to be cost-efficient for dwellings exposed only to floods with a return period higher than a hundred years. Our results apply to the whole France. Decision-makers could rely on them to recommend precautionary measures only to inhabitants that live in dwellings for which they could be advantageous.

*Code availability.* *floodam* is developed as a R library. It is available upon request through a git repository maintained by Irstea, and under certain conditions.

*Data availability.* Data at building levels used within *floodam* are available upon request through a git repository maintained by Irstea, and under certain conditions.

5  *Author contributions.* Claire Richert modelled the elevation strategy, analysed the data, and wrote the article. Hélène Boisgontier modelled the other strategies. Frédéric Grelot developed *floodam*, supervised the project, and proofread the article.

*Competing interests.* The authors declare that they have no conflict of interest.

*Acknowledgements.* This work benefited from the support of the French Ministry of Environment, through the DGPR program "Knowledge and prevention of natural and hydraulic hazards" (funding decision number 2102049246).

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

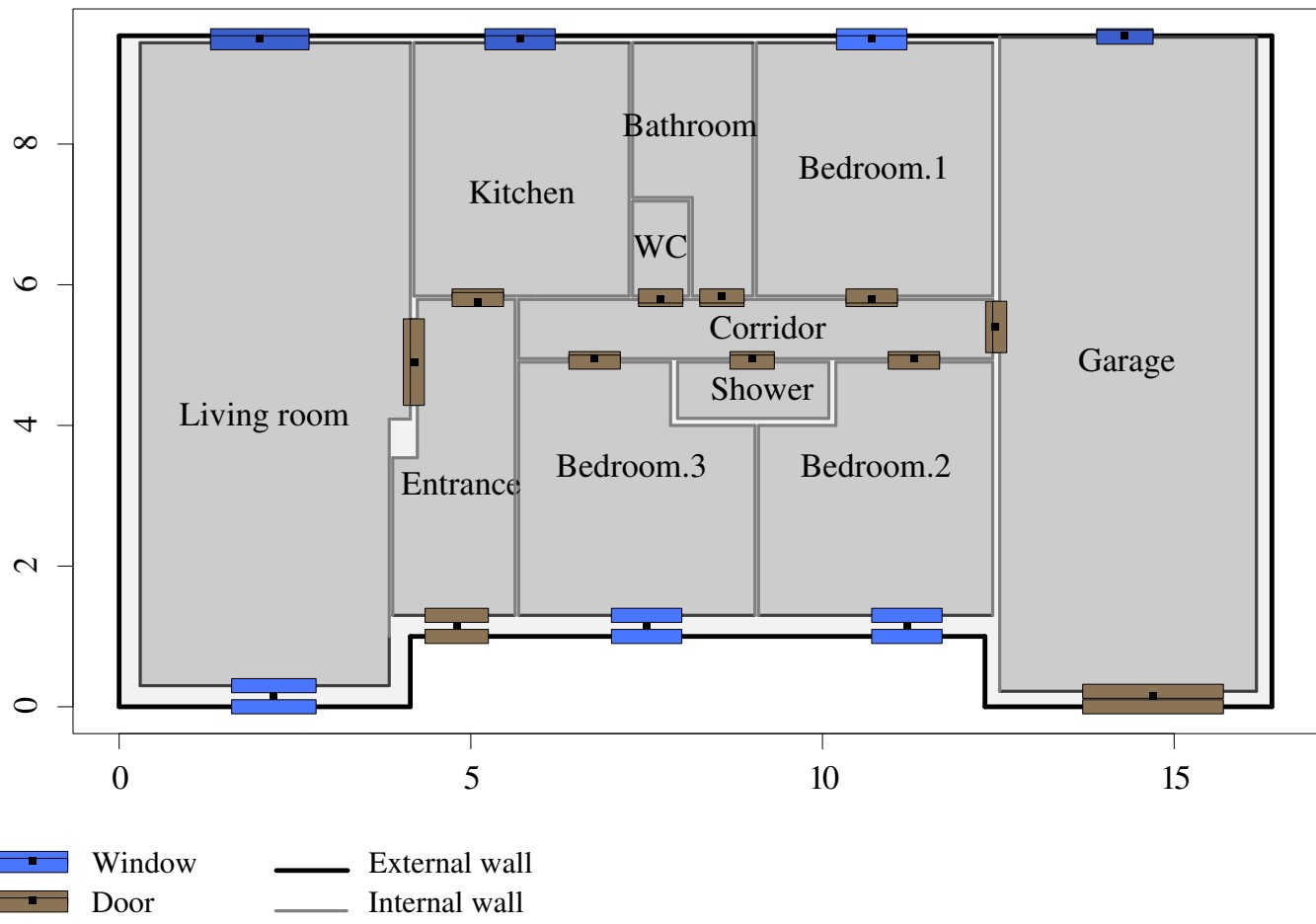

**Figure 1.** Top view of the single storey house. The external walls are in black, the internal walls in grey, the doors in brown, and the windows in blue.

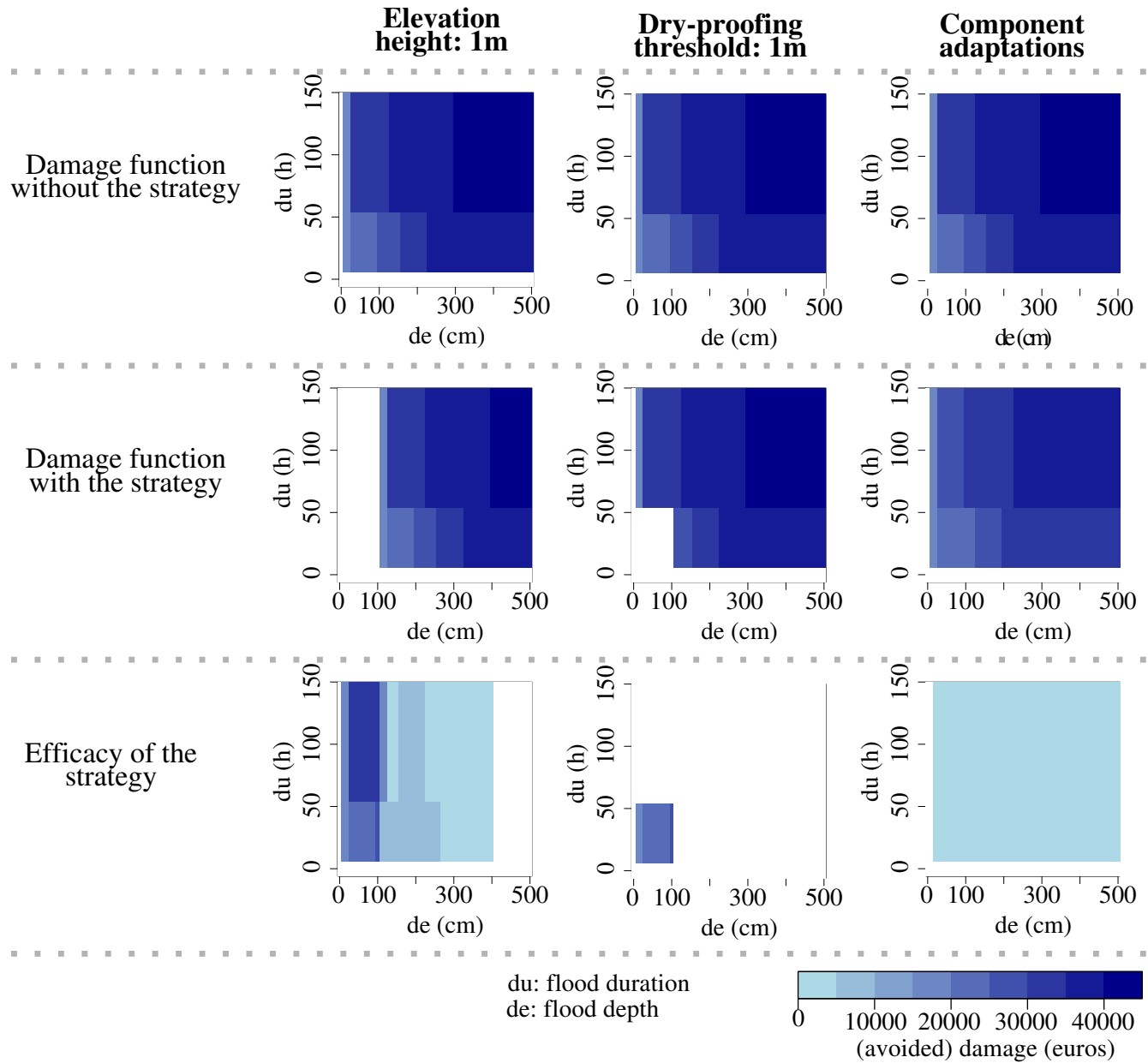

**Figure 2.** Efficacy of the elevation, dry-proofing and component adaptations strategies implemented on the original version of the single storey house. The two top panels show damage in euros while the lower panel shows avoided damage in euros.

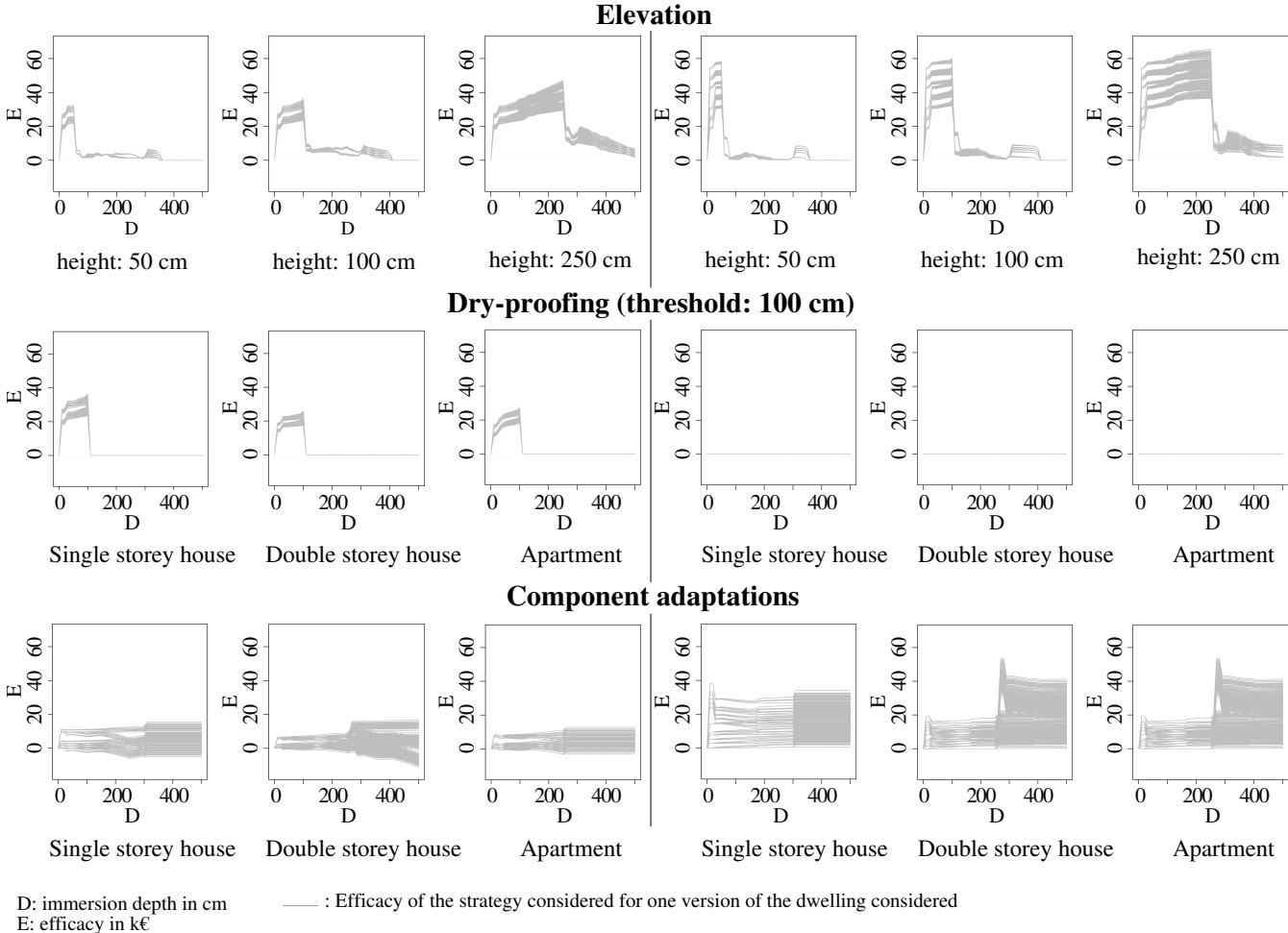

D: immersion depth in cm
E: efficacy in k€

———— : Efficacy of the strategy considered for one version of the dwelling considered

**Figure 3.** Efficacy of the three strategies for a duration of immersion of 24 or 72 hours

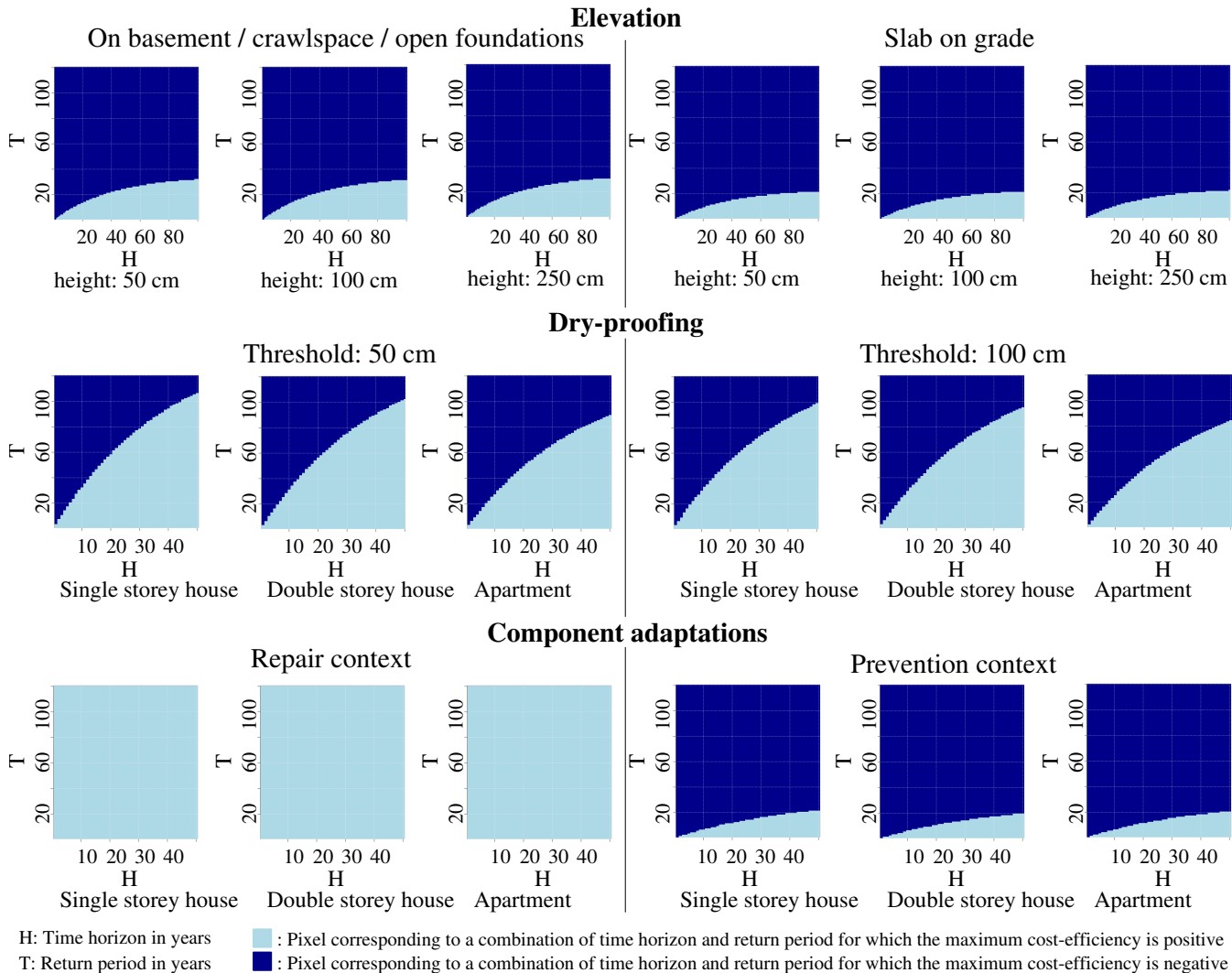

**Figure 4.** Sign of the maximum cost-efficiency of each strategy

**Table 1.** Materials that limit the costs of repair or replacement after a flood

| Component | Material |
|---|---|
| Ceilings | Plaster boards and metal frame |
| Insulation of ceilings | Cellular plastic |
| Internal walls | Plaster boards and metal frame |
| Insulation of walls | Cellular plastic |
| Floors | Concrete |
| Coatings of floors | Sealed tiling |
| Internal door frames | Metal |
| External door frames | PVC or metal |
| Window frames | PVC |
| Shutters | PVC |

**Table 2.** Pieces of furniture contained in Bedroom.1 of the single storey house (see Fig. 1)

| Item | Height above the floor (cm) | Quantity |
|---|---|---|
| Single bed in solid wood | 0 | 1 |
| Stock of linen (value: €2675) | 0 | 1 |
| Various furniture in chipboard | 0 | 3 |
| Small items (value: €1500) | 0 | 0.3 |
| Small items (value: €1500) | 120 | 0.7 |
| Stock of toys (value: €900) | 120 | 1 |

**Table 3.** Components of the building used to develop the versions of the numerical models of dwellings

| Component | Number of variants | variant |
|---|---|---|
| External walls | 3 | brick, reinforced concrete, concrete blocks |
| External walls render | 1 | Cement |
| Internal walls | 2 | plaster boards/metal frame, masonry |
| Insulation | 2 | mineral wool, plastic |
| Coatings of walls | 2 | paint, paper |
| Ceilings | 3 | wood planks and frame, plaster boards/metal frame, concrete |
| Coatings of ceilings | 1 | paint |
| Floors | 2 | concrete, joist board |
| Coatings of floors | 3 | tiles, parquet, textile |
| Opening frames | 2 | PVC, wood |
| Shutters | 2 | PVC, wood |

**Table 4.** Costs of elevating a masonry dwelling as reported by FEMA (2009)

| Type of foundations | Height (feet) | Cost (US$/foot$^2$) |
|---|---|---|
| Basement / crawlspace / open | 2 | 60 |
| Basement / crawlspace / open | 4 | 63 |
| Slab on grade | 2 | 88 |
| Slab on grade | 4 | 91 |

The costs by square foot are in 2009 US$

**Table 5.** Costs of elevating a masonry dwelling used in our study

| Type of foundations | Height (cm) | Cost (€/m$^2$) |
|---|---|---|
| Basement / crawlspace / open | 50 | 497 |
| Basement / crawlspace / open | 100 | 517 |
| Basement / crawlspace / open | 250 | 579 |
| Slab on grade | 50 | 731 |
| Slab on grade | 100 | 751 |
| Slab on grade | 250 | 813 |

The costs by square meter are in 2017 €

**Table 6.** Costs of the dry-proofing measures

| Measure | Cost by measurement unit | Quantity |
|---|---|---|
| Flood barriers | 775 €/unit | N of O |
| Removable covers | 15.5 €/m | P |
| Repair of faulty seals | 35 €/m$^2$ | P × T |
| Treatment of cracks | 3 €/m$^2$ | P × T |
| Waterproofing of the walls | 30 €/m$^2$ | P × T |
| Watertight electrical sleeves | €550 | NA |
| Pumping device | €820 | NA |
| Anti-backflow valves | €650 | NA |

N of 0: Number of openings; P: Perimeter; T: threshold; NA: we assume that the cost of the measure is fixed.

**Table 7.** Distribution of the cost in k€ across all numerical models of dwellings

| Strategy | Case | Minimum | Mean | Maximum | Standard deviation |
|---|---|---:|---:|---:|---:|
| Elevation (On basement / crawlspace / open foundations) | 50 cm | 66.3 | 66.3 | 66.3 | 0.0 |
| | 100 cm | 69.1 | 69.1 | 69.1 | 0.0 |
| | 250 cm | 77.3 | 77.3 | 77.3 | 0.0 |
| Elevation (Slab on grade) | 50 cm | 97.5 | 97.5 | 97.5 | 0.0 |
| | 100 cm | 100.3 | 100.3 | 100.3 | 0.0 |
| | 250 cm | 108.5 | 108.5 | 108.5 | 0.0 |
| Dry-proofing (Threshold: 50 cm) | Single storey house | 8.6 | 8.6 | 8.6 | 0.0 |
| | Double storey house | 6.3 | 6.3 | 6.3 | 0.0 |
| | Apartment | 7.6 | 7.6 | 7.6 | 0.0 |
| Dry-proofing (Threshold: 100 cm) | Single storey house | 10.4 | 10.4 | 10.4 | 0.0 |
| | Double storey house | 7.7 | 7.7 | 7.7 | 0.0 |
| | Apartment | 9.2 | 9.2 | 9.2 | 0.0 |
| Component adaptations (Repair context) | Single storey house | -33.2 | -12.6 | 8 | 9.9 |
| | Double storey house | -45.5 | -18.2 | 8.9 | 13.1 |
| | Apartment | -28.4 | -11.4 | 5.5 | 7.7 |
| Component adaptations (Prevention context) | Single storey house | 2.6 | 40.2 | 74.0 | 17.2 |
| | Double storey house | 1.8 | 53.9 | 102.2 | 24.9 |
| | Apartment | 0.3 | 30.0 | 57.1 | 13.4 |

**Table 8.** Distribution of efficacy in k€ across all numerical models of dwellings and all combinations of immersion depth and duration

| Strategy | Case | Minimum | Mean | Maximum | Standard deviation |
|---|---|---|---|---|---|
| Elevation | 50 cm | 0.0 | 4.5 | 58.3 | 10.1 |
| | 100 cm | 0.0 | 9.0 | 60.1 | 13.8 |
| | 250 cm | 0.0 | 22.1 | 65.4 | 18.2 |
| Dry-proofing | Single storey house | 0.0 | 0.7 | 32.3 | 4.2 |
| (Threshold: 50 cm) | Double storey house | 0.0 | 0.5 | 22.9 | 3.1 |
| | Apartment | 0.0 | 0.5 | 23.9 | 2.8 |
| Dry-proofing | Single storey house | 0.0 | 1.6 | 36.3 | 6.2 |
| (Threshold: 100 cm) | Double storey house | 0.0 | 1.1 | 25.9 | 4.5 |
| | Apartment | 0.0 | 1.1 | 27.4 | 4.5 |
| Component adaptations | Single storey house | -8.9 | 9.5 | 38.3 | 8.9 |
| | Double storey house | -13.3 | 8.8 | 53.6 | 10.0 |
| | Apartment | -5.6 | 7.6 | 26.7 | 6.5 |