# Peer review of "Economic assessment of precautionary measures against floods: insights from a non contextual approach"

_Natural Hazards and Earth System Sciences, 2019_

## Referee Comment (RC1) · Anonymous Referee #1 · 9 Jul 2019

The articles assess the economic benefits of household level measures to reduce flood damage. The article is well written, clear, the methods seem solid and carried out very well. My only concern is with the innovation compared to earlier similar studies. As the article points out many similar studies have been conducted and reached similar conclusions. This study is new in that it is using more sophisticated synthetic damage functions and possibly a slightly different method to estimate the annual expected efficacy (not entirely clear).

- Point out even better what is new and innovative compared to earlier studies. This should become clear from the abstract and the introduction and maybe even the title.

- I don't like the word "computer modelling" in the title. Almost everything is computer modelling nowadays. Can you find another word to describe what makes this article new compared to the earlier studies

- You look at a maximum duration of 144 hours. Some floods can last many months, could you discuss this choice and give an estimate of how the conclusions would differ for longer flood durations.

Minor comments

Page 1, line 17: You talk about 2 components (hazard and vulnerability) but exposure is a third component (even though irrelevant when looking at only an individual building.

Page 5, line 16: "model of dwelling" should I think be "model of a dwelling"

Page 6, line 12: You look at a maximum duration of 144 hours. Some floods can last many months

Page 6, from line 25: Not entirely sure what you mean by perimeter.

Page 9, formula 5,6 and 7: Could you use a comma between sd, now it looks like an undefined variable rather than s and d.

Page 9: Can you provide an intuitive explanation of maximum cost-efficiency.

Page 13, line 18: "flood barriers must be completed by other measures", I think you mean complemented instead of completed.

Throughout the paper: Can you provide price levels whenever a monetary value is mentioned.

---

## Referee Comment (RC2) · Anonymous Referee #2 · 11 Jul 2019

This study adopts a synthetic flood loss model to assess the efficacy and cost-efficiency of three types of precautionary measures of dwellings. The article is generally well written and there is some content of interest here to the flood loss modelling and risk community. However, in my opinion the article requires a complete overhaul before it can be considered publishable.

General comments:

Cost-benefit analyses are one of the fundamental reasons for performing risk assessment studies, and have been carried out for years by all types of stakeholders involved in disaster risk reduction (e.g. governmental bodies, industry, academia). I do not see

anything particularly innovative from a scientific viewpoint in Section 3.5. As such, the novelty of the article and the approach is assumed to relate to the application of a new synthetic loss model called floodam to support such analyses.

In this context, it should be noted that the scientific field of flood loss modelling has received increasing attention in the last couple of decades, and the body of literature has become vast. The introduction of this article, which attempts to explain why it is relevant, does not engage with much of the relevant literature on this topic. There are both empirical and synthetic models which use precautionary measures or other building properties to explain flood loss, and can therefore be used to make similar analyses. See for example Gerl et al., 2016; Sairam et al., 2019. This does not mean that there is no scope to propose new flood loss models – on the contrary. Flood loss modelling remains an open field where more research is certainly warranted. However, the reader should be able to understand what is the novelty of this research, which is currently not the case. This is the main shortcoming of your article.

Since the model is based on a synthetic approach, this modeling approach should be given particular attention in your literature review and in positioning your new model among it. I can think of at least three articles proposing synthetic flood loss models comparable to this one, i.e. where losses are obtained based on losses to individual building components: Custer and Nishijima, 2015; Dottori et al., 2016; Nadal et al., 2010. The general description presented in Section 3.3. of your article could apply almost word by word to any of these models. In light of this, I would suggest you restructure the article such that much more emphasis is given to the new flood loss model, and that you then illustrate its application through the economic analyses performed here. From the user manual of floodam, it appears to be a well-structured model that deserves to be presented to the scientific community following peer-review.

Specific comments:

Given the extensive revision that I think the article requires, I am only making two

specific comments at this point. The first is related with the calculation of the annual expected efficacy (AEE). In P2-L8 the authors correctly state the AAE can be obtained through a probability weighted average of the values of efficacy for different flood intensities. However, it is not clear in the article how the authors have actually calculated this. The probability of exceeding certain flood intensity measures such as water depth depends on the asset location, and as such, the average annual losses are site-dependent. Therefore, the cost-efficiency of precautionary measure is necessarily also site-dependent. What was considered here? This should be clarified in a new version of the manuscript. The second comment relates to the applicability of your findings. I assume these are meant to apply to France, but no explicit reference is made to this aspect in the article. In the abstract you also mention some findings (e.g. "according to our results, dry-proofing and elevating a dwelling are unlikely to be cost-efficient for dwellings that are not exposed to floods with a return period lower than 100 and 30 years, respectively") but a reference to where they are assumed to be valid is missing.

References:

Custer, R., and Nishijima, K. (2015). Flood vulnerability assessment of residential buildings by explicit damage process modelling. Natural Hazards (Vol. 78). Springer Netherlands. doi:10.1007/s11069-015-1725-7

Dottori, F., Figueiredo, R., Martina, M., Molinari, D., and Scorzini, A. R. (2016). IN-SYDE: a synthetic, probabilistic flood damage model based on explicit cost analysis. Natural Hazards and Earth System Sciences, 16(12), 2577–2591. doi:10.5194/nhess-16-2577-2016

Gerl, T., Kreibich, H., Franco, G., Marechal, D., and Schröter, K. (2016). A Review of Flood Loss Models as Basis for Harmonization and Benchmarking. Plos One, 11(7). doi:10.1371/journal.pone.0159791

Nadal, N. C., Zapata, R. E., Pagán, I., López, R., and Agudelo, J. (2010). Building Damage due to Riverine and Coastal Floods. Journal of Water Resources Planning

and Management, 136(3), 327–336. doi:10.1061/(ASCE)WR.1943-5452.0000036

Sairam, N., Schröter, K., Lüdtke, S., Merz, B., and Kreibich, H. (2019). Quantifying Flood Vulnerability Reduction via Private Precaution. Earth's Future, 7(3), 235–249. doi:10.1029/2018EF000994

---

## Author Comment (AC1) · 26 Jul 2019

Dear Referee,

Thank you for your comments.

Regarding your general concern about the novelty of our article, we agree that we should emphasize it better in the abstract and in the introduction.

We think that the novelty of our article lies both in our research question and in our results.

We explain in the following why we think that our research question is original.

[Figure]

The purpose of the other articles that deal with the assessment of precautionary measures is mainly to examine their mean efficacy (or cost-effiency) in specific contexts. By contrast, we aim to present in-depth analyses of how these efficacy and cost-efficiency vary depending on the buildings characteristics and exposure to floods. Such analyses are useful in order to better target the dwellings for which precautionary measures could be advantageous.

We also think that we provide new and useful results. Mainly, we present a systematic methodology to identify the conditions (in terms of exposure to riverine floods) in which the measures cannot be cost-efficient, no matter the building materials. We apply this methodology to identify these conditions for the whole France. This result can be used by decision-makers to recommend precautionary measures only to inhabitants that live in dwellings for which precautionary measures could be advantageous.

Your second comment was: "Point out even better what is new and innovative compared to earlier studies. This should become clear from the abstract and the introduction and maybe even the title."

Thank you for this comment.

We propose to change the title to: "Economic assessment of precautionary measures against floods: insights from a non contextual approach"

In the abstract, we propose to replace the 2 following sentences "In particular, a better understanding of the influence of buildings characteristics and floods parameters on these variables would help identify the measures that are relevant to implement in specific contexts. We examined this topic for three groups of measures taken on existing dwellings: [...]" by: "In particular, the influence of buildings and floods characteristics on these variables has not been thoroughly studied. A better understanding of this topic would help identify the measures that are relevant to implement in specific contexts. To address this gap, we examined the effect of buildings and floods characteristics on the cost, efficacy, and cost-efficiency of three groups of measures taken on existing

dwellings: [. . .]."

In the introduction, we propose to modify the paragraph on page 3, lines 12-16 by: "In brief, the existing literature focuses on assessing the efficacy or cost-efficiency of precautionary measures, rather than on explaining their variability. The aim of our study was to address this gap. We combined data based on expert judgement and computer modelling to analyse three types of measures (elevation, dry-proofing, and components adaptations) for a wide range of flood intensities and dwellings characteristics, including the material used for their components. More specifically, we assessed ranges of cost and efficacy of the measures and examined the influence of buildings and floods characteristics on these variables. For each type of measures, we also found a range of exposure level for which it is unlikely that the measure could be cost-efficient, independently of the buildings characteristics."

Your third comment was: "I don't like the word "computer modelling" in the title. Almost everything is computer modelling nowadays. Can you find another word to describe what makes this article new compared to the earlier studies"

We propose the following title: "Economic assessment of precautionary measures against floods: insights from a non contextual approach"

Your fourth comment was: "You look at a maximum duration of 144 hours. Some floods can last many months, could you discuss this choice and give an estimate of how the conclusions would differ for longer flood durations."

Thank you for this comment. We propose to discuss this point in Section "5.4 Limits":

"We only studied the efficacy and cost-efficiency of the strategies for dwellings that are exposed to floods that do not last more than 144 hours because the experts interviewed to develop the elementary damage functions did not have information about the consequences of longer floods. The efficacy of dry-proofing for such floods is null because it is recommended to let the water enter the building after 48 hours. As for the

elevation strategy, its efficacy in case of floods that last more than 144 hours depends on the propensity of such floods to generate foundations failure. If the foundations fail, the fact that the building is elevated does not reduce the damage. The efficacy of the component adaptations strategy for floods longer than 144 hours depends on the vulnerability of the recommended components when they are in contact with water for more than 144 hours."

Minor comments:

We agree with your three first minor comments and will take them into account in the article.

Your fourth minor comment was: "Page 6, from line 25: Not entirely sure what you mean by perimeter."

We mean: the length of the boundary of the building. Maybe "circumference" is more appropriate?

We also agree with your fifth comment.

Your sixth comment was: "Page 9: Can you provide an intuitive explanation of maximum cost-efficiency?"

The maximum cost-efficiency is a supremum of the cost-efficiency. In other words, for a given strategy and a given type of building (single storey house, double storey house, or apartment), the cost-efficiency of the strategy is always lower than the maximum cost-efficiency, regardless of the building materials or the relationship between the flood intensity and frequency.

Your seventh comment was: "Page 13, line 18: "flood barriers must be completed by other measures", I think youmean complemented instead of completed."

Thank you for pointing this out. You are right.

We did not understand your last comment: "Throughout the paper: Can you provide

price levels whenever a monetary value is mentioned." Could you provide an example?

---

## Author Comment (AC2) · 26 Jul 2019

Dear Referee,

Thank you for your comments and for the references you suggested. We reply to each of your remark below.

General comments:

In the first paragraph, you point out that the novelty of our article relates to the application of a new synthetic model to assess the cost-efficiency of several precautionary measures.

We think that the novelty of our article lies both in our research question and in our results. However, we agree that the current version of our article does not emphasize it enough and that we should explain it more clearly in the abstract, introduction, and conclusion.

We explain in the following why we think that our research question is original.

The purpose of the other articles that deal with the assessment of precautionary measures is mainly to examine their mean efficacy (or cost-effiency) in specific contexts. By contrast, we aim to present in-depth analyses of how these efficacy and cost-efficiency vary depending on the buildings characteristics and exposure to floods. Such analyses are useful in order to better target the dwellings for which precautionary measures could be advantageous.

We also think that we provide new and useful results. Mainly, we present a systematic methodology to identify the conditions (in terms of exposure to riverine floods) in which the measures cannot be cost-efficient, no matter the building materials. We apply this methodology to identify these conditions for the whole France. This result can be used by decision-makers to recommend precautionary measures only to inhabitants that live in dwellings for which precautionary measures could be advantageous.

In the second paragraph, you underline that we should refer to empirical and synthetic models which use precautionary measures or other building properties to explain flood loss, and that could therefore be used to make similar analyses.

We agree that we should compare floodam to existing flood loss models in order to explain why it seems to be the most relevant to conduct our analyses. Thank you for suggesting some references. We propose to add the following section, called "Suitability of floodam to assess the precautionary measures" after "3.3 Overview of floodam":

"Numerous empirical and synthetic flood loss models exist. Floodam belongs to the latter category. Empirical models are based on observed flood loss data, whereas
synthetic models rely on a description of flood damage mechanisms (Gerl et al., 2016).

Some empirical flood damage models include precautionary measures as explanatory variables (see for example Kreibich et al. (2017)) and can be used to estimate their mean efficacy (Sairam et al., 2019). They account for the mean effect of all measures in the sample used to produce them on flood damage. Thus, they cannot be used to estimate the efficacy of specific precautionary measures. Moreover, the influence of the flood parameters and building characteristics on the efficacy of precautionary measures cannot be deduced from these models.

The damage mechanisms are more explicit in synthetic models (see for example Custer and Nishijima (2015), Dottori et al. (2016), Nadal et al. (2010), and Zevenbergen et al. (2007)). They can be altered to depict the effect of specific precautionary measures. Floodam has one main advantage over the other existing synthetic models to assess the efficacy of precautionary measures. To our knowledge, no other model is based on such detailed database of elementary damage functions. This enables us to examine the influence of a wide variety of building materials on the efficacy of specific precautionary measures."

We should also specify in section "3.3 Overview of floodam" that floodam is used to estimate the damage due to floods that do not cause failure of walls and that do not involve salt water.

Moreover, we should compare our results with those of Zevenbergen et al. (2007) in Section "5.2 Comparison with previous studies".

In the third paragraph, you suggest to restructure the article to give more emphasis to floodam and present our analysis as an application of this model.

In our view, floodam is only a tool to analyse the cost-efficiency of some precautionary measures. The aim of the paper (that we should state more clearly in the introduction) is really 1) to examine how the building characteristics and flood parameters influence
the efficacy and cost of the measures and 2) to identify exposure levels for which the measures cannot be cost-efficient. Floodam is thoroughly described in its manual, which is free, publicly available, and written in English. However, in the article, we agree that we should more clearly explain why floodam is more suitable than the other flood loss models to examine the cost-efficiency of precautionary measures. As explained above, we intend to add a section after Section 3.3 to do so.

Specific comments: Your first comment is the following: "The first is related with the calculation of the annualexpected efficacy (AEE). In P2-L8 the authors correctly state the AEE can be obtained through a probability weighted average of the values of efficacy for different flood intensities. However, it is not clear in the article how the authors have actually calculated this. The probability of exceeding certain flood intensity measures such as water depth depends on the asset location, and as such, the average annual losses are site-dependent. Therefore, the cost-efficiency of precautionary measure is necessarily also site-dependent. What was considered here?"

Thank you for this comment. We agree that our explanation was not very clear. Indeed, the cost-efficiency of a precautionary measure is necessarily site-dependent. However, we wanted to find exposure levels for which each measure is not cost-efficient in general, independently of the flood intensity linked to these exposure levels. Thus, we found a supremum of cost-efficiency (that we call "maximum cost-efficiency") for each measure and each type of building (single storey house, double storey house, apartment). This supremum is not site-dependent because it is not calculated with the Annual Expected Efficacy, but with a supremum of the Annual Expected Efficacy. This supremum is the ratio between 1) the maximum efficacy of the measure over all possible combinations of water depth and submersion duration, and over all combinations of building materials, and 2) the return period of the flood that affects the dwelling the most often. The first term is constant and thus, not site-dependent. The second term is not fixed. It is the one we are interested in: we explored its influence on the maximum cost-efficiency to find exposure levels for which the maximum cost-efficiency is
negative. In these cases, according to our results, the measure will always be costinefficient, regardless of the building materials and the relationship between the flood intensity and flood frequency, since the maximum cost-efficiency is already negative.

In order to explain this better, we propose to add at the end of Section 3.5.3: "Indeed, unlike the cost-efficiency, the maximum cost-efficiency does not depend on the building materials and on the relationship between the flood parameters (water depth and immersion duration) and frequency. It only depends on the time horizon and return period. Thus, when the maximum cost-efficiency is negative, it means that the strategy will always be cost-inefficient, regardless of the building materials and on the relation-ship between the flood parameters and frequency."

We also propose to make equation (4) more explicit: "AEE = int\_{0}^{1} f x E(de(f), du(f)) df" with de the water depth and du the flood duration.

Your second comment is the following: "The second comment relates to the applicability of your findings. I assume these are meant to apply to France, but no explicit reference is made to this aspect in the article. In the abstract you also mention some findings (e.g. "according to our results, dry-proofing and elevating a dwelling are unlikely to be cost-efficient for dwellings that are not exposed to floods with a return period lower than 100 and 30 years, respectively") but a reference to where they are assumed to be valid is missing."

Thank you for pointing this out. You are right. We should add the following sentence at the end of the abstract: "Our results apply to France because the damage and the cost of the measures are specific to France and the geometry of the dwellings considered to perform our analyses is based on French dwellings."

We should also add a sentence about this issue in Section "5.4 Limits" and in the conclusion.

References:
Custer, R., & Nishijima, K. (2015). Flood vulnerability assessment of residential buildings by explicit damage process modelling. Natural Hazards, 78(1), 461–496. https://doi.org/10.1007/s11069-015-1725-7

Dottori, F., Figueiredo, R., Martina, M. L. V, Molinari, D., Scorzini, A. R., Civile, I., & Milano, P. (2016). INSYDE : a synthetic , probabilistic flood damage model based on explicit cost analysis, 2577–2591. https://doi.org/10.5194/nhess-16-2577-2016

Gerl, T., Kreibich, H., Franco, G., Marechal, D., & Schröter, K. (2016). A Review of Flood Loss Models as Basis for Harmonization and Benchmarking. PLoS ONE, 11(7), 1–22. https://doi.org/10.1371/journal.pone.0159791

Kreibich, H., Botto, A., Merz, B., & Schröter, K. (2017). Probabilistic , Multivariable Flood Loss Modeling on the Mesoscale with BT-FLEMO. Risk Analysis, 37(4), 774–787. https://doi.org/10.1111/risa.12650

Nadal, N. C., Zapata, R. E., Pagán, I., López, R., & Agudelo, J. (2010). Building Damage due to Riverine and Coastal Floods. Journal of Water Resources Planning and Management, 136(3), 327–336. https://doi.org/10.1061/(ASCE)WR.1943-5452.0000036

Sairam, N., Schröter, K., Lüdtke, S., Merz, B., & Kreibich, H. (2019). Quantifying Flood Vulnerability Reduction via Private Precaution. Earth's Future, 7(3), 235–249. https://doi.org/10.1029/2018EF000994

Zevenbergen, C., Gersonius, B., Puyan, N., & Van Herk, S. (2007). Economic Feasibility Study of Flood Proofing Domestic Dwellings. In C. Ashley, Richard and Garvin, Stephen L. and Pasche, Erik and Vassipoulos, Andreas and Zevenbergen (Ed.), Advances in urban flood management (pp. 299–319). CRC Press. https://doi.org/10.1201/9780203945988

---

## Author Response (AR1)

**A) Comments from referees/public**

**I/ Referee 1**

Major comments:

1) "Point out even better what is new and innovative compared to earlier studies. This should become clear from the abstract and the introduction and maybe even the title."

2) "I don't like the word "computer modelling" in the title. Almost everything is computer modelling nowadays. Can you find another word to describe what makes this article new compared to earlier studies?"

3) "You look at a maximum duration of 144 hours. Some floods can last many months, could you discuss this choice and give an estimate of how the conclusions would differ for longer flood durations?"

Minor comments:

1) "Page 1, line 17: You talk about 2 components (hazard and vulnerability) but exposure is a third component (even though irrelevant when looking at only an individual building)."

2) "Page 5, line 16: "model of dwelling" should I think be "model of a dwelling"."

3) "Page 6, line 12: You look at a maximum duration of 144 hours. Some floods can last many months."

4) "Page 6, from line 25: Not entirely sure what you mean by perimeter."

5) "Page 9, formula 5, 6 and 7: Could you use a comma between sd, now it looks like an undefined variable rather than s and d."

6) "Page 9: Can you provide an intuitive explanation of maximum cost-efficiency?"

7) "Page 13, line 18: "flood barriers must be completed by other measures", I think you mean complemented instead of completed."

8) "Throughout the paper: Can you provide price levels whenever a monetary value is mentioned?"

**II/ Referee 2**

Major comments:

1) "Cost-benefit analyses are one of the fundamental reasons for performing risk assessment studies, and have been carried out for years by all types of stakeholders involved in disaster risk reduction (e.g. governmental bodies, industry, academia). I do not see anything particularly innovative from a scientific viewpoint in Section 3.5. As such, the novelty of the article and the approach is assumed to relate to the application of a new synthetic loss model called floodam to support such analyses."

2) "It should be noted that the scientific field of flood loss modelling has received increasing attention in the last couple of decades, and the body of literature has become vast. The introduction

of this article, which attempts to explain why it is relevant, does not engage with much of the relevant literature on this topic. There are both empirical and synthetic models which use precautionary measures or other building properties to explain flood loss, and can therefore be used to make similar analyses. See for example Gerl et al., 2016; Sairam et al., 2019. This does not mean that there is no scope to propose new flood loss models – on the contrary. Flood loss modelling remains an open field where more research is certainly warranted. However,the reader should be able to understand what is the novelty of this research, which is currently not the case. This is the main shortcoming of your article."

3) "Since the model is based on a synthetic approach, this modelling approach should be given particular attention in your literature review and in positioning your new model among it. I can think of at least three articles proposing synthetic flood loss models comparable to this one, i.e. where losses are obtained based on losses to individual building components: Custer and Nishijima, 2015; Dottori et al., 2016; Nadal et al.,2010. The general description presented in Section 3.3. of your article could apply almost word by word to any of these models. In light of this, I would suggest you restructure the article such that much more emphasis is given to the new flood loss model, and that you then illustrate its application through the economic analyses performed here. From the user manual of floodam, it appears to be a well-structured model that deserves to be presented to the scientific community following peer-review."

Minor comments:

1) "In P2-L8 the authors correctly state the AAE can be obtained through a probability weighted average of the values of efficacy for different flood intensities. However, it is not clear in the article how the authors have actually calculated this. The probability of exceeding certain flood intensity measures such as water depth depends on the asset location, and as such, the average annual losses are site-dependent. Therefore, the cost-efficiency of precautionary measure is necessarily also site-dependent. What was considered here? This should be clarified in a new version of the manuscript."

2) "I assume these are meant to apply to France, but no explicit reference is made to this aspect in the article. In the abstract you also mention some findings (e.g. "according to our results, dry-proofing and elevating a dwelling are unlikely to be cost-efficient for dwellings that are not exposed to floods with a return period lower than 100 and 30 years, respectively") but a reference to where they are assumed to be valid is missing."

**III/ Additional comment of the editor**

1) "Also the applicability of the scientific findings needs to be emphasized and demonstrated more in depth."

**B) Authors' response**

**I/ Response to referee 1**

Response to the major comments

1) In order to point out what is new in our study, we provided the following changes:

- We changed the title to: "Economic assessment of precautionary measures against floods: insights from a non contextual approach". Indeed, the main novelty of our study lies in the fact that we examined the cost, efficacy, and cost-efficacy of precautionary measures in a non contextual setting. This enabled us to explore the influence of several building characteristics and flood parameters on the cost and efficacy of the measures.

- In the abstract, we replaced these 2 sentences: "In particular, a better understanding of the influence of buildings characteristics and floods parameters on these variables would help identify the measures that are relevant to implement in specific contexts. We examined this topic for three groups of measures taken on existing dwellings: [...]" by: "In particular, the influence of building and flood characteristics on these variables has not been thoroughly studied.  A better understanding of this topic would help identify the measures that are relevant to implement in specific contexts. To address this gap, we examined the effect of building and flood characteristics on the cost, efficacy, and cost-efficiency of three groups of measures taken on existing dwellings: […]" (page 1, lines 3-6)

- In the introduction, we modified the paragraph on page 3, lines 14-20 by: "In brief, the existing literature focuses on assessing the efficacy or cost-efficiency of precautionary measures, rather than on explaining their variability. The aim of our study was to address this gap. We combined data based on expert judgement and computer modelling to analyse three types of measures (elevation, dry-proofing, and components adaptations) for a wide range of flood intensities and dwellings characteristics, including the material used for their components. More specifically, we assessed ranges of cost and efficacy of the measures and examined the influence of building and flood characteristics on these variables. For each type of measures, we also found a range of exposure level for which it is unlikely that the measure could be cost-efficient, independently of the buildings characteristics."

2) We changed the title to: "Economic assessment of precautionary measures against floods: insights from a non contextual approach"

3) In the new version of the manuscript, we discuss our choice to look at a maximum flood duration of 144 hours in Section "5.4 Limits", page 15, lines 21-28: "We only studied the efficacy and cost-efficiency of the strategies for dwellings that are exposed to floods that do not last more than 144 hours because the experts interviewed to develop the elementary damage functions did not have information about the consequences of longer floods. The efficacy of dry-proofing for such floods is null because it is recommended to let the water enter the building after 48 hours. As for the elevation strategy, its efficacy in case of floods that last more than 144 hours depends on the propensity of such floods to generate foundations failure. If the foundations fail, the fact that the building is elevated does not reduce the damage. The efficacy of the component adaptations strategy for floods longer than 144 hours depends on the vulnerability of the recommended components when they are in contact with water for more than 144 hours."

Response to the minor comments:

1) We changed the two first sentences of the introduction (page 1, lines 18-19) to: "A flood risk can be defined as the combination of a hazard, an exposed population, and its vulnerability to the hazard."

2) We wrote "model of a dwelling" instead of "model of dwelling" for each occurrence of these terms.

3) We added the following footnote on page 6, line 12: "Some floods can last more than 144 hours. The experts interviewed did not provide data regarding the vulnerability of the elementary components to such floods. Thus, we did not study the efficacy of precautionary measures for floods longer than 144 hours."

4) By perimeter, we mean: the length of the boundary of the building. Maybe "circumference" is more appropriate? Since we are not sure, we did not change this term.

5) We used a comma between s and d in the equations 5, 6, and 7, pages 9 and 10.

6) On page 9, lines 2-6, we added the following explanation of the maximum cost-efficiency: "It is thus a supremum of the cost-efficiency for the type of dwelling considered. In other words, for a given strategy and a given type of dwelling, the cost-efficiency of the strategy is always lower than the maximum cost-efficiency, regardless of the building materials or the relationship between the flood intensity and frequency."

7) On page 14, line 2, we replaced "completed" by "complemented".

8) We apologize but we did not understand this comment. Could you give us an example of what you mean by "price levels"?

**II/ Response to referee 2**

Response to the major comments:

1) We think that the novelty of our article lies both in our research question and in our results.

The purpose of the other articles that deal with the assessment of precautionary measures is mainly to examine their mean efficacy (or cost-effiency) in specific contexts. By contrast, we aim to present in-depth analyses of how these efficacy and cost-efficiency vary depending on the buildings characteristics and exposure to floods. Such analyses are useful in order to better target the dwellings for which precautionary measures could be advantageous.

We also think that we provide new and useful results. Mainly, we present a systematic methodology to identify the conditions (in terms of exposure to riverine floods) in which the measures cannot be cost-efficient, no matter the building materials. We apply this methodology to identify these conditions for the whole France. This result can be used by decision-makers to recommend precautionary measures only to inhabitants that live in dwellings for which precautionary measures could be advantageous.

We modified the manuscript in order to emphasize the novelty of our study:
- In the abstract, we replaced these 2 sentences: "In particular, a better understanding of the influence of buildings characteristics and floods parameters on these variables would help identify the measures that are relevant to implement in specific contexts. We examined this topic for three groups of measures taken on existing dwellings: [...]" by: "In particular, the

influence of building and flood characteristics on these variables has not been thoroughly studied. A better understanding of this topic would help identify the measures that are relevant to implement in specific contexts. To address this gap, we examined the effect of building and flood characteristics on the cost, efficacy, and cost-efficiency of three groups of measures taken on existing dwellings: […]" (page 1, lines 3-6)

- In the introduction, we modified the paragraph on page 3, lines 14-20 by: "In brief, the existing literature focuses on assessing the efficacy or cost-efficiency of precautionary measures, rather than on explaining their variability. The aim of our study was to address this gap. We combined data based on expert judgement and computer modelling to analyse three types of measures (elevation, dry-proofing, and components adaptations) for a wide range of flood intensities and dwellings characteristics, including the material used for their components. More specifically, we assessed ranges of cost and efficacy of the measures and examined the influence of building and flood characteristics on these variables. For each type of measures, we also found a range of exposure level for which it is unlikely that the measure could be cost-efficient, independently of the buildings characteristics."

- In the conclusion, we replaced the first sentence by: "We analysed three types of precautionary measures in a non contextualized setting. This novel approach enabled us to explore the influence of several building and flood characteristics on the cost and efficacy of the precautionary measures and to find ranges of exposure for which they are unlikely to be cost-efficient." (Page 15, lines 30-32).

2) We compared floodam to existing flood loss models in order to explain why it seems to be the most relevant to conduct our analyses in a new section, from page 6, line 22 to page 7 line 3:

"3.4 Suitability of floodam to assess the precautionary measures

Numerous empirical and synthetic flood loss models exist. Floodam belongs to the latter category. Empirical models are based on observed flood loss data, whereas synthetic models rely on a description of flood damage mechanisms (Gerl et al., 2016).

Some empirical flood damage models include precautionary measures as explanatory variables (see for example Kreibich et al. (2017)) and can be used to estimate their mean efficacy (Sairam et al., 2019). These models account for the mean effect on flood damage of all the measures observed in the case studies used to produce them. Thus, they cannot be used to estimate the efficacy of specific precautionary measures. Moreover, the influence of the flood parameters and building characteristics on the efficacy of precautionary measures cannot be deduced from these models.

The damage mechanisms are more explicit in synthetic models (see for example Custer and Nishijima (2015), Dottori et al. (2016), Nadal et al. (2010), and Zevenbergen et al. (2007)). They can be altered to depict the effect of specific precautionary measures. Floodam has one main advantage over the other existing synthetic models to assess the efficacy of precautionary measures. To our knowledge, no other model is based on such detailed database of elementary damage functions. This enables us to examine the influence of a wide variety of building materials on the efficacy of specific precautionary measures."

3) In our view, floodam is only a tool to analyse the cost-efficiency of some precautionary measures. The aim of the paper (that we tried to state more clearly in the abstract, introduction, and conclusion) is really 1) to examine how the building characteristics and flood parameters influence the efficacy and cost of the measures and 2) to identify exposure levels for which the measures

cannot be cost-efficient. Floodam is thoroughly described in its manual, which is free, publicly available, and written in English.

However, in the new version of the manuscript, we explained why floodam is more suitable than the other flood loss models to examine the cost-efficiency of precautionary measures in the new section "3.4. Suitability of floodam to assess the precautionary measures".

Response to the minor comments:

1) The maximum cost-efficiency is a supremum of the cost-efficiency for a given strategy and a given type of building (single storey house, double storey house, apartment). This supremum is not site-dependent because it is not calculated with the Annual Expected Efficacy, but with a supremum of the Annual Expected Efficacy. This supremum is the ratio between 1) the maximum efficacy of the measure over all possible combinations of water depth and submersion duration, and over all combinations of building materials, and 2) the return period of the flood that affects the dwelling the most often. The first term is constant and thus, not site-dependent. The second term is not fixed. It is the one we are interested in: we explored its influence on the maximum cost-efficiency to find exposure levels for which the maximum cost-efficiency is negative. In these cases, according to our results, the measure will always be cost-inefficient, regardless of the building materials and the relationship between the flood intensity and flood frequency, since the maximum cost-efficiency is already negative.
In order to better explain this point:

- On page 9, lines 2-6, we added: "It is thus a supremum of the cost-efficiency for the type of dwelling considered. In other words, for a given strategy and a given type of dwelling, the cost-efficiency of the strategy is always lower than the maximum cost-efficiency, regardless of the building materials or the relationship between the flood intensity and frequency."

- At the end of Section "3.6.3 Assessment of the maximum cost-efficiency", we added: "Indeed, unlike the cost-efficiency, the maximum cost-efficiency for a given type of dwelling does not depend on the building materials and on the relationship between the flood intensity (immersion depth and duration) and frequency. It only depends on the time horizon and return period. Thus, for the combinations of time horizon and return period associated to a negative maximum cost-efficiency, the strategy will always be cost-inefficient, regardless of the building materials and on the relationship between the flood intensity and frequency." (page 10, lines 12-16)

- We made it explicit in equation 4 that the Annual Expected Efficacy depends on the relationship between the flood intensity and frequency through the efficacy term, so that it is clearer in equation 5 and 6 that the maximum cost efficiency does not depend on this relationship (because the efficacy term is fixed).

2) To make it explicit that our results apply to France:
- We added the following sentence at the end of the abstract: "Our results apply to France because the damage and the installation costs of the measures are specific to France and the geometry of the dwellings considered to perform our analyses is based on French dwellings." (Page 1, lines 15-16)

- We added the following paragraph in Section "5.4 Limits" (page 15, lines 18-20): "Our results apply to France. To conduct our study in another country, we should adapt the cost of the measures, the cost of the actions needed after a flood to go back to the initial state (which are used to estimate the damage), and the geometry of the dwellings."

**III/ Response to the additional comment of the editor**

1) In order to emphasize the applicability of our findings:

- We added a sentence in Section "5.3 Recommendations based on our results" (page 15, lines 4-5): "Moreover, policy-makers should not promote the installation of dry-proofing measures in dwellings that are not exposed to floods with a return period lower than 100 years."

- We added the following sentence in the conclusion (page 16, lines 3-4: "Decision-makers could rely on [our results] to recommend precautionary measures inly 
[revised manuscript text omitted]

---

## Author Response (AR2)

**A) Comments from Referee 2**

General comment

"My main remaining criticism continues to relate to how the floodam model is presented, and its role within this article. While I agree that floodam is "only a tool to analyse the cost-efficiency of some precautionary measures" (quoting the authors), the results that are presented obviously depend on the assumptions and structure of the model that is adopted. Therefore, I believe this is a critical point of the article that deserves particular attention.

Although floodam does have a well structured English manual, this manual does not appear to go into too much detail, putting more emphasis on the general methodology. The model code itself is not publicly available, and as far as I understand floodam has not been validated based on real data or peer-reviewed. Also, P6 L8 states that floodam is based on 431 elementary functions. This level of detail is then presented as an advantage over other models. However, in my opinion this is a somewhat unsupported statement, because the model is not available for review and no proof is given of the predictive skill of these functions or the model as a whole.

To conclude: I suppose floodam has limitations - as does any model - and I also suppose it does not address all the unresolved issues in flood loss modelling. If that is the case, I believe this is something that should be discussed in the article, possibly in Section 5.4. In particular, how can possible model limitations affect the conclusions that the authors have drawn in the article regarding these precautionary measures?"

Minor comments

1) "P1 L20: Exposure does not relate to population exclusively, but to any element at risk. Please correct accordingly."

2) "P9 L15: I found the use of "contextualized dwelling" a bit confusing here. For clarity, please add an explanation of what this means."

**B) Authors' response**

General comment

- We reformulated our statement of P6 L8 to make clear that we do not think the advantage of floodam lies in its predictive skills, but in its ability to disaggregate the damage function of a dwelling into numerous elementary damage functions:

  "To our knowledge, no other model is based on such detailed database of elementary damage functions. This characteristic enabled us to examine the influence of a wide variety of building materials on the efficacy of specific precautionary measures."

- We reorganized Sect. 5.4 ("Limits") into two subsections: "Limits linked to the study perimeter" and "Limits linked to the method". In this latter, we:
  - make clear that some limits previously mentioned are linked to the assumptions and data on which floodam relies.
  - added a paragraph where we mention that some work has already been done to check the validity of floodam and highlight that further work is needed to validate the damage functions used in our study. We underline that validating these damage functions would require very detailed data regarding a flood event. At this time, we do not have such data.

"floodam is the tool used by the French State to produce currently recommended national flood damage functions to be used in cost-benefit analyses. The comparison of these damage functions to formerly recommended empirical flood damage functions and to empirical data collected in the South of France (CEPRI, 2014) led to this recommendation. The national damage functions were also used in a case study in the South and France and the estimates obtained were compared with empirical flood damage data (Richert and Grelot, 2018): the mean damage estimate amounted to 70% (99%) of the mean empirical damage to houses (apartments). Nevertheless, in the present article, we have used floodam for a wide range of configurations for which validation against empirical data has not been done. Such validation would imply to have access to detailed data about a given flood: they should indicate precisely the location and characteristics (in terms of building materials, furniture, and geometry) of the dwellings of the flooded area, the flood depth and duration outside and inside these dwellings, and the flood damage suffered by each dwelling. At the present moment, this type of data are not available."

Minor comments

1) P1 L19-20: We replaced the sentence by: "A flood risk can be defined as the combination of a hazard, exposed assets and populations, and their vulnerability to the hazard (e.g. Apel et al. (2009))"

[revised manuscript text omitted]